# dagLogo: An R/Bioconductor package for identifying and visualizing differential amino acid group usage in proteomics data

Jianhong Ou[1,2☯], Haibo Liu[1☯], Niraj K. Nirala[3], Alexey Stukalov[4], Usha Acharya[1¤], Michael R. Green[1], Lihua Julie Zhu[1,3,5]*

1 Department of Molecular, Cell, and Cancer Biology, University of Massachusetts Medical School, Worcester, Massachusetts, United States of America, 2 Regeneration NEXT, Duke University School of Medicine, Duke University, Durham, North Carolina, United States of America, 3 Program in Molecular Medicine, University of Massachusetts Medical School, Worcester, Massachusetts, United States of America, 4 Institute of Virology, Technical University of Munich, Munich, Germany, 5 Program in Bioinformatics and Integrative Biology, University of Massachusetts Medical School, Worcester, Massachusetts, United States of America

☯ These authors contributed equally to this work.
¤ Current address: Cancer and Developmental Biology Laboratory, National Cancer Institute, Frederick, Maryland, United States of America
* Julie.Zhu@umassmed.edu

**Data Availability Statement:** All relevant data are within the manuscript and its Supporting Information files.

## Abstract

Sequence logos have been widely used as graphical representations of conserved nucleic acid and protein motifs. Due to the complexity of the amino acid (AA) alphabet, rich post-translational modification, and diverse subcellular localization of proteins, few versatile tools are available for effective identification and visualization of protein motifs. In addition, various reduced AA alphabets based on physicochemical, structural, or functional properties have been valuable in the study of protein alignment, folding, structure prediction, and evolution. However, there is lack of tools for applying reduced AA alphabets to the identification and visualization of statistically significant motifs. To fill this gap, we developed an R/Bioconductor package dagLogo, which has several advantages over existing tools. First, dagLogo allows various formats for input sets and provides comprehensive options to build optimal background models. It implements different reduced AA alphabets to group AAs of similar properties. Furthermore, dagLogo provides statistical and visual solutions for differential AA (or AA group) usage analysis of both large and small data sets. Case studies showed that dagLogo can better identify and visualize conserved protein sequence patterns from different types of inputs and can potentially reveal the biological patterns that could be missed by other logo generators.

## Introduction

Since their first introduction in 1990, sequence logos have been widely used to visualize conserved patterns among large sets of nucleic acid and peptide sequences [1–4]. Dozens of sequence logo generators with different functionalities and performances have been

**Funding:** The author(s) received no specific funding for this work.

**Competing interests:** The authors have declared that no competing interests exist.

developed; these generators require various types of inputs and background models, and use different algorithms and graphical representations [1–3, 5–15]. A comprehensive list of these tools, along with their features, is available in S1 Table. Almost all tools provide no alignment utilities but require multiple sequence alignments in various formats, or position weight/frequency matrices as inputs. A majority of these tools are also based on information theory without statistical significance tests [16–18]. More than two thirds of the existing tools cannot effectively display both over- and under-represented residues (S1 Table), even though the information on under-represented residues may be as valuable as that of over-represented residues [12, 19].

It is known that both background models and sizes of input sets affect the identification and visualization of sequence patterns [3, 7, 18, 20]. However, a majority of logo generators do not allow users to choose a dataset-specific background model. For those that do allow it, the only option for a majority of them is a background model based on global residue frequencies or GC%, which can be specified by users, calculated from input sequences, or derived from a species-specific whole genome or proteome (S1 Table). These simple background models usually work well for identifying and visualizing nucleic acid patterns due to the simple nucleotide alphabet and limited modifications of nucleic acids, especially DNA. However, in many cases, they may not be optimal for protein pattern discovery because of the complexity of the amino acid (AA) alphabet, rich post-translational modification of proteins, diverse subcellular localization, cell or tissue type-specific expression profiles, and experimental protocol-specific biases. To our knowledge, only Two Sample Logo [13], iceLogo [7, 21], pLogo [6], kpLogo [22], and PTM-Logo [23] offer some advanced options of background models. Among them, iceLogo offers the most choices, i.e., i) species-specific, proteome-wide AA frequencies as the background; ii) position-specific AA frequencies as the background; and iii) an experimental protocol bias-aware background.

To date, there are no tools that have the functionalities to group AAs based on their physicochemical or other properties and test the differential usage of AA groups. Historically, twenty naturally occurring AAs have been classified into two to 19 groups based on some measures of their relative similarity in physicochemical, structural, and functional properties; this has resulted in a variety of reduced (degenerate or simplified) alphabets [24–27]. With reduced alphabets, several AAs of a similar property are lumped together and represented by one letter. Reduced AA alphabets have been shown to be valuable for protein pattern discovery [28–30] and in the study of protein alignment, folding, structure prediction, and evolution [25, 31–38]. For example, serine (Ser) and threonine (Thr) are highly interchangeable as phosphorylation sites of Ser/Thr kinase substrates [39], and thus the two AAs can be binned into the same degenerate group to reduce the complexity of resulting logos showing conserved patterns of Ser/Thr kinase (STK) substrates. Thus, unsurprisingly, coloring AA residues based on their similarity of physicochemical properties has been a general practice of many logo generators long before [2] (S1 Table). However, applying reduced AA alphabets to motif identification and visualization had rarely been practiced until recent [28, 29]. This limitation is more obvious for sequence patterns of proteins than sequence patterns of nucleic acids because protein sequences have a greater alphabet complexity and variability of the occurrences of individual residues. Even though RaacLogo [28] and logoJS [29] allow degenerate representations of reduced AA alphabets [40, 41], both were based on information theory without statistical significance tests, and most importantly, neither offers alternative background models (S1 Table).

Taken together, none of the existing tools has implemented all the following functionalities: allowing flexible formats for input sets, providing comprehensive options to build optimal background models, and implementing reduced AA alphabets to group AAs of similar properties. To fill this gap, we have developed an open source R/Bioconductor package dagLogo for

identifying and visualizing conserved peptide patterns with the probability theory. In addition to having the functionality to generate various background models, dagLogo also accepts aligned subsequences or unaligned subsequences of different lengths as input, and it visualizes significantly over- and under-represented AA residues or AA residue groups in experimental context-aware ways. We demonstrate here that dagLogo can better identify and visualize conserved patterns hidden in peptide sequences regardless of length and alignment status, and can potentially reveal patterns that could be missed by other logo generators.

## Materials and methods

### Implementation

dagLogo is implemented as an open source R/Bioconductor [42] package. Several S4 classes are implemented to represent different datatypes: Class *dagPeptides* for peptide sequences, Class *Proteome* for proteomes, Class *dagBackground* for background models, and Class *testDAUresults* for statistical test results of differential usage of AA residues/groups. Functions implemented in the dagLogo package are described in S2 Table and the package vignette.

### Workflow of dagLogo analyses

A flowchart for a typical dagLogo-based analysis is shown in Fig 1. A dagLogo analysis can start with creating a *Proteome* object, which stores protein identifiers (IDs), protein sequences, species information, and the source of sequences. One can prepare the *Proteome* object with the *prepareProtome* function by providing a fasta file containing the proteome sequences of the species. Alternatively, the user can specify the species' scientific name or NCBI taxonomy ID with the same function, which will automatically download the species-specific proteome data from the UniProt database.

Next, a *dagPeptides* object representing a formatted input set can be created in several ways. For pre-aligned input subsequences, one can use the *formatSequence* function with the specified *proteome* object, as well as upstream and downstream offsets relative to the anchoring positions of the input subsequences. If pre-aligned input subsequences are unavailable, the *dagPeptides* object can be created by using the *fetchSequence* function in four different ways by providing IDs of protein sequences from which input subsequences originate, the options to access full-length protein sequences from an Ensembl Biomart database or a *proteome* object, the anchoring AA(s) and the anchoring AA's positions. Prior to fetching sequences, input data can be cleaned up with the *cleanPeptides* function, which can remove peptides without anchoring AAs or duplicated peptides, and can generate an individual entry for each anchoring AA if an original entry contains multiple anchoring AAs.

Like iceLogo, dagLogo requires a background model, which is derived from background subsequences of the same length as those in the input set, to properly calculate the probability of the number of occurrences of given AAs at each position in the input set. Choosing an appropriate background model is therefore of great importance, and the background set should suit the actual technical and biological contexts. Twelve background models can be generated for *Z*-test or Fisher's exact test (see below) using the *buildBackgroundModel* function by specifying i) the proteome-level background-generating space as the whole proteome ("WholeProteome"), proteins corresponding to the input set ("InputSet"), or the whole proteome excluding those proteins in "InputSet" ("NonInputSet"); ii) the protein-level background-generating space as N-termini only, *C*-termini only, or anywhere of full-length proteins; iii) anchoring AAs or not when the protein-level background-generating space is not restricted to protein termini. A proper background model should be considered to meet experimental and analytical needs [7]. UniProt, Biomart, and/or anchored AAs are leveraged to fetch the peptide

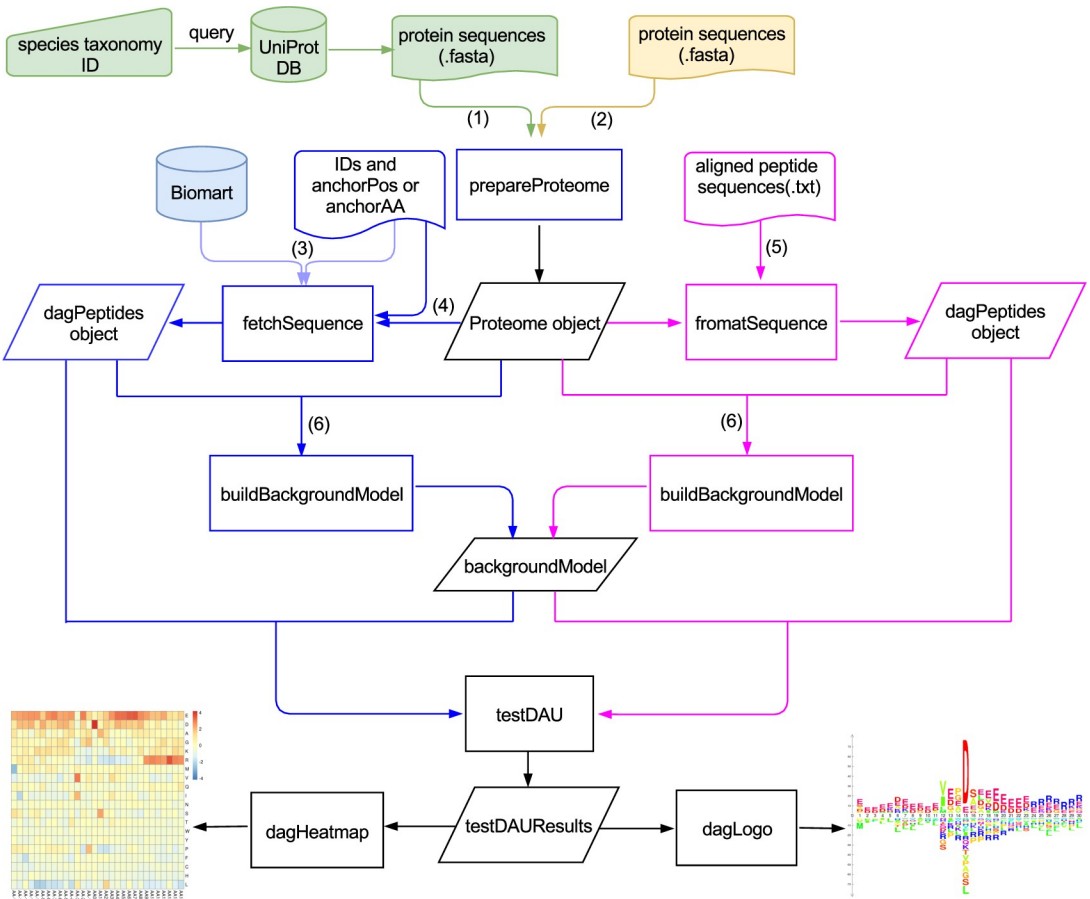

**Fig 1. dagLogo workflow.** When whole proteome sequences in the fasta format are not available for a species, the NCBI taxonomy ID or scientific name of the species is needed to query the UniProt database to obtain the desired fasta file (1) and to build a Proteome object using the prepareProteomeByFTP or prepareProteomeByUniprotws function. These two functions are wrapped into the prepareProteome function. Otherwise, an existing fasta file (2) is directly used to build a Proteome object with the prepareProteome function. A dagPeptides object for an input set can be constructed in three ways with unaligned (i and ii) subsequences in different lengths or aligned (iii) subsequences: i) When a pre-aligned sequence set is not available, Entrez gene IDs or UniProt SwissProt protein IDs, as well as anchoring AAs (such as "K" for Lys) or anchoring AA position information of the input set, can be used to obtain sequences from the species-specific Biomart database for creating a dagPeptides object using the fetchSequence function. An anchoring AA within a peptide sequence is represented by a lowercase single-letter symbol or an uppercase single-letter symbol followed by an asterisk, e.g., the anchoring AA Lys in "ASTRSkSSTD" or "ASTRSK*SSTD". Meanwhile, an anchoring AA position is represented as a string consisting of the anchoring AA followed by the AA's position, such as "K123" for the 123th residue—Lys (3); ii) With the same input information as the previous method, a dagPeptides object can be built from the species-specific Proteome object rather than the Biomart database using the fetchSequence function (4); iii) If the peptide sequences of the input set have already been aligned, the dagPeptides object can be built from the species-specific Proteome object using the formatSequence function. Twelve different background models for Fisher's exact test or Z-test can be built from the Proteome object using the buildBackground function with different parameter settings (6). A differential AA or AA group usage (DAU) analysis for the input set (the dagPetides object) can be performed using the testDAU function, with the appropriate background model represented by a backgroundModel object). Heatmaps and logos generated by the functions dagHeatmap and dagLogo, respectively, can be used to visualize the results of the DAU analysis.

sequences, which have characteristics or constraints similar to those of the experimental data, to build a specific background model. For example, if an experiment only focuses on the mitochondrial proteome, then an appropriate background model would be built from the entire or a subset of the mitochondrial proteome that is anchored to the AA(s) of interest. If the experiment is to identify peptides with acetylated lysine, then an appropriate background model would be derived from all peptide fragments containing anchoring lysine.

Once the type of background model to build is decided, a Fisher's exact test or *Z*-test approximation can be applied to identify significantly enriched/depleted AAs or AA groups. For Fisher's exact test, a single background set with all available background subsequences will be generated. If both the input set's size and the number of subsequences for the background set are large enough, *Z*-test approximation may be preferred, as it can speed up the tests of differential AA or AA group usage (DAU). To do so, bootstrapping is used to generate multiple background sets that conform to the background-generating rules, with each background set consisting of the same number of equal-length sequences as those in the input set. The averaged frequency ($\overline{p_l^{AA}}$) of a given residue *AA* at position *l* and the corresponding standard error $\overline{s_l^{AA}}$ are calculated as follows:

$$\overline{p_l^{AA}} = \sum_{t=1}^{T} p_{lt}^{AA}/T \tag{1}$$

$$\overline{s_l^{AA}} = \sqrt{\overline{p_l^{AA}}(1 - \overline{p_l^{AA}})/N} \tag{2}$$

where $p_{lt}^{AA}$ is the frequency of residue *AA* at position *l* for subsample *t*, with $l \in [1,L]$ and $t \in [1, T]$. *L* is the length of the pattern, *T* is the total number of subsamples of background sets, and *N* is the number of subsequences in each subsample. Notably, the number of sequences in each background set and the total number of background sets can affect the accuracy of the estimates for AA frequencies and their standard errors. A normal distribution $N(\overline{p_l^{AA}}, \overline{s_l^{AA}})$ is used to approximate the distribution of the proportion of a given residue at each position.

Next, given the background model built as above, the significance of the proportion of an AA at a given pattern position in the experimental set is tested by using the *testDAU* function. Performing Fisher's exact test is straightforward. To perform a two-sided *Z*-test, we assume the frequency of a residue at a given pattern position of the input set approximately follows the normal distribution $N(\overline{p_l^{AA}}, \overline{s_l^{AA}})$, which is inferred from the background sets for Z-test. A Z-score is calculated as follows:

$$Z = (p_l^{AA} - \overline{p_l^{AA}})/\overline{s_l^{AA}} \tag{3}$$

where $p_l^{AA}$ is the frequency of a given *AA* at given position *l* in the input set. The test result is an object of Class *testDAUresults*. The test results can be visualized as a heatmap using the *dagHeatmap* function, which leverages the *pheatmap* package. Significantly over- and under-represented residues can be visualized as sequence logos using the dagLogo function, which is built on the grid graphics system in R.

Importantly, the package also provides an array of grouping schemes to represent AA residues of similar properties with degenerate, single-letter symbols. AAs can be grouped according to the indices of their isoelectric point (pI) [43], polarity [44], hydrophobicity [45, 46], bulkiness [43], volume [47], and consensus similarity [48] as individual AAs or substitutability within protein contexts [49–51]. In addition, dagLogo allows users to specify customized grouping and coloring schemes through the *addScheme* function. Users can refer to the AAindex database [40] or other resources [26, 52] to create their own grouping schemes. With grouped AAs represented by reduced AA alphabets, dagLogo can improve the detection power of identifying significant degenerate residues and can generate degenerate sequence logos.

## Datasets for case studies

We demonstrated the applications of our dagLogo package with three datasets as described below. The UniProt release 2020_05 was used in this study.

**Dataset 1: Substrates of the human GRB.** We downloaded the S1 Table by Van Damme *et al.* [53], and made the following modifications: i) four obsolete UniProtKB-TrEMBL identifiers (IDs) were changed to the current UniProt Swiss-Prot IDs of proteins with identical sequences, as determined by BLAST analyses; ii) IDs of three proteins with accession numbers starting with "IPI" were changed to their current UniProt Swiss-Prot IDs; and iii) six substrates without current Swiss-Prot records (O55106, O95365, P78559, Q00839, Q06210 and Q149F1) were removed. These manipulations resulted in 416 human protein substrates of the human GRB for the case study. The processed data are included as S3 Table. Equal-length subsequences (P15-P15') centered on the human GRB cutting sites—the peptide bonds between P1-P1'—were prepared using the *fetchSequence* function in the dagLogo package. Herein P15 to P1 are residues at positions 15 to 1 N-terminal to the cutting sites, while P1' to P15' are residues at position 1 to 15 *C*-terminal to the cutting sites. The letter X was used to pad the peptide sequences when the fetched sequence's length was shorter than 30. These 416 subsequences were used as the input set to build a dagPeptides object. The background set consisted of the 300 subsets of 416 subsequences of 30 AA residues randomly sampled from the UniProt human reference proteome (https://www.uniprot.org/proteomes/UP000005640).

**Dataset 2: Substrates of human Ser/Thr protein kinases.** We downloaded phosphorylated peptide sequences from all known kinase substrates from the PhosphoSitePlus database (https://www.phosphosite.org/) (file name: "Kinase_Substrate_Dataset.gz"). Then, we extracted the UniProt IDs, peptide sequences of 15 AAs (phosphorylation site ± 7 AAs), and absolute positions of phosphorylation sites of 867, 646, and 651 substrates of the human kinases PKACA, PKCA and CK2A1 from 11 species, including the human. Of these sequences, 469, 385, and 465 are human substrates of the human kinases PKACA, PKCA, and CK2A1, respectively. The data are included as S4 Table. To determine the human substrate sequence specificities of each human kinase, the same background model was built using randomly sampled subsequences of 15 AAs from the UniProt human reference proteome (https://www.uniprot.org/proteomes/UP000005640). To compare the substrate sequence preference of one kinase with that of another, multi-species substrate subsequences of 15 AAs (phosphorylation site ± 7 AAs) of one kinase were used as the input set, and the multi-species substrate subsequences of 15 AAs (phosphorylation site ± 7 AAs) of another kinase were used as the background set.

**Dataset 3: Substrates of the yeast N-terminal acetyltransferase A.** The latest baker's yeast reference proteome was downloaded from the UniProt database (https://www.uniprot.org/proteomes/UP000002311). The N-terminal Met residue was removed from each protein sequence in the entire proteome for subsequent analysis for two reasons: i) the first Met residue is cleaved co-translationally *in vivo* if the next residue is glycine, alanine, serine, threonine, cysteine, proline, or valine [54]; ii) all eukaryotic NatA does not acetylate a peptide or protein with a Met N-terminus [55–58]. The data used for mining the substrate sequence specificity of yeast NatA was extracted from the S3 Table with the following modifications [56]. Obsolete IDs of six substrates (P04451, P26781, P26782, P35271, P40213, and P53030) were updated to the current UniProt Swiss-Prot IDs. Misinformation of the mature N-terminal 5-AA peptide sequences (column name: "mature_N_term_P5") of 11 substrates (Q12074, P02829, P00360, P38720, P04076, P15019, P10591, Q03048, P53880, P38088, and P35187) were corrected based on the yeast reference proteome. With these manipulations, the final data consisted of information about 285 substrates of the yeast NatA. Peptide sequences consisting of the first 25-AA

of the N-termini were retrieved from the yeast proteome using the *fetchSequence* function in the dagLogo package. The data are included as S5 Table. Different background sets (see Results) were adopted depending on the analysis purposes.

## Results

### Case study 1: dagLogo successfully identifies the substrate sequence specificity of granzyme B at both individual AA and physicochemical property levels

Granzyme B (GRB) is an apoptotic serine protease found in secretory granules of cytotoxic T lymphocytes and natural killer cells [59, 60]. Early *in vitro* screening studies revealed that GRB preferentially cleaves the peptide bonds immediately proceeded by aspartate residues at P1 positions [61], and determinants of its substrate specificity span from residues at P4 to P2' (for notation Px and Px', see Fig 2) [62]. Harris *et al.* identified an optimal consensus substrate of GRB with (I/V)-(E/Q/M)-X-(DX)-G at positions P4 to P2', where X is any AA residue and the peptide bond between D and X is the GRB cleavage site [62]. Later, an *in vivo* study identified the proteome-wide substrates for the human as well as the mouse GRB [53], which not only confirmed the *in vitro* findings about the determinants of GRB substrate specificity but also revealed that more residues from P3' to P9' are related to substrate specificities and cutting efficiency [53]. Van Damme *et al.* further revealed the physicochemical property constraints in the substrate sequence pattern in terms of charge, hydrophobicity, and side chain size of residues [53]. GRB prefers negatively charged AAs at P3 and P2'-P8', and slightly dislikes positively charged AAs around the cleavage site. They also found that AAs at P4 tend to be hydrophobic, followed by less hydrophobic AA at P3. Additionally, AAs at positions P3, P2, and P1' flanking the cleavage site tend to be small residues [53]. These observations are consistent with the molecular structure of the GRB substrate binding pocket [63–65].

We reanalyzed the proteome-wide substrate data of the human GRB [53] using our dagLogo package and compared the resulting logo with logos generated by other common tools with the same input and background sets of subsequences, wherever possible. S1 Fig displays the sequence patterns among the human GRB substrates revealed by dagLogo, iceLogo, pLogo, and WebLogo. Although all logo generators successfully identified the major determinant of substrate specificity, i.e., enriched Asp in P1, both pLogo and WebLogo failed to reveal the important determinants flanking P1 (S1C and S1D Fig). pLogo calculates the log odds, log $(p/(1-p))$, where $p$ is the binomial probability of an AA residue occurring at a given position. pLogo emphasizes the big differences but downplays the minor differences. This may explain why residues flanking P1, which contributes less to substrate specificity, are not as clearly represented. WebLogo does not allow users to provide a background model. As a result, a uniform background frequency of 20 AA residues was used as the background model by default. In addition, it can only display positively scored residues based on the information theory. Logos generated by dagLogo (S1A Fig) and iceLogo (S1B Fig) are very similar, which clearly displays both the major and minor determinants of the substrate specificity. The subtle differences might be because the two tools calculate the variances differently (see Discussion).

Next, we performed more analyses of the human GRB substrate specificity, using dagLogo to display other important features of our package. Fig 2A displays the differential usage of each AA residue from P15 to P15' through a heatmap. This heatmap gives a detailed view of AA distribution at each position in comparison with the given background model. Fig 2B, 2C, 2E and 2G display significant differential usages of individual AAs at each position, where AA residues are colored individually or collectively according to their hydrophobicity indices, pI indices, and sizes, respectively. By contrast, Fig 2D, 2F and 2H show significant differential

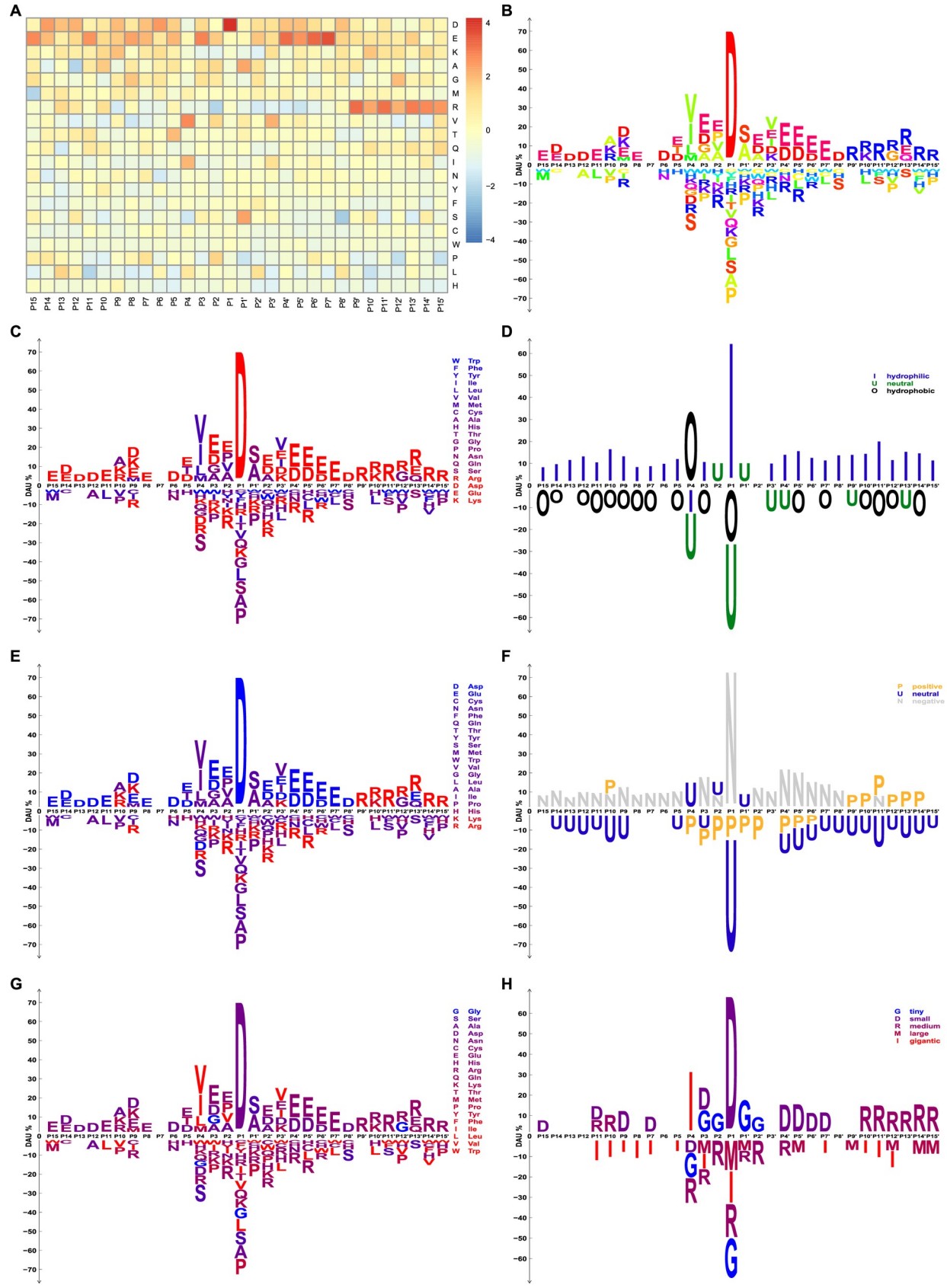

**Fig 2. Heatmap and sequence logos showing the substrate sequence pattern of the human granzyme B.** A background model for *Z*-test was built from randomly sampled subsequences of 30 AA residues from the UniProt human reference proteome. The significance level is set as 0.05 for DAU tests. (A) *Z*-scores of DAU tests for each residue at each pattern position are shown in the heatmap. (B, C, E, and G) Over- and under-represented AA residues with significant DAU at each pattern position are shown as sequence logos. Individual AA residues are colored differently and according to their hydrophobicity indices, pI indices, or sizes, respectively. (D, F, and H) AA residues are grouped and collapsed based on their hydrophobicity indices, charge status under physiological conditions, and sizes, respectively. Over- and under-represented AA residue groups with significant DAU at each pattern position are shown as degenerate logos. AA residue groups are colored according to their hydrophobicity indices: (D) hydrophobic AAs = {W, F, Y, L, I, V, M, C}, neutral AAs = {H, A, T, P, G, N, A, S}, and hydrophilic AAs = {R, K, D, E}; their charge status under physiological conditions: (F) positively charged AAs = {H, K, R}, neutral AAs = {A, C, F, G, I, L, M, N, P, Q, S, T, V, W, Y}, and negatively charged AAs = {D, E}; or sizes: (H) tiny = {G, S, A}, small = {D, N, C, E, H}, medium = {R, Q, K, T}, large = {M, P, Y, F}, and gigantic = {I, L, V, W}. Van Damme *et al.* proposed that the weak arginine conservation from P11' to P15' is probably due to technical artifacts caused by the N-terminal COFRADIC sorting procedure [53].

usages of AA groups using degenerate logos, where AAs sharing a similar physicochemical property are grouped using the reduced AA alphabets and are tested for differential usage. dagLogo-based analyses, as displayed in Fig 2, recapitulated the previous *in vitro* and *in vivo* findings. These findings not only confirm the properties of GRB substrate specificity in terms of sequence preferences at individual AA level, but also illuminate the restrictions in physical properties like hydrophobicity, charge, and sizes of residues [53, 61, 62].

## Case study 2: dagLogo successfully differentiates substrate specificities of three human Ser/Thr protein kinases

One of the most common, important, and dynamically reversible post-translation modifications is phosphorylation, which is controlled by various protein kinases and phosphatases [66–68]. Protein kinases employ combinatorial mechanisms to define their substrate specificities, including substrate peptide specificity and substrate recruitment specificity [69, 70]. Substrate motifs of protein kinases have been identified over the past 70 years, though different studies have resulted in more or less different forms of motifs [71–75] (https://www.phosphosite.org). According to the AA residues phosphorylated by protein kinases, eukaryotic kinases are classified into Ser/Thr kinases (STKs), Tyr kinases, and dual-specificity kinases, which can phosphorylate Ser, Thr, and Tyr residues. At least 125 of the 568 human protein kinases are STKs [76, 77], and more than 86% of phosphorylated residues are Ser, followed by 11.8% being Thr and 1.8% being Tyr in the human HeLa cells [78].

PKACA (cAMP-dependent protein kinase catalytic subunit alpha) and PKCA (protein kinase C alpha) are members of the AGC protein kinase superfamily. The consensus substrate motif of PKACA is $(R/K)(R/K)X(S^*/T^*)$, where $S^*/T^*$ is the phosphorylated Ser or Thr [79, 80] (https://www.phosphosite.org). It prefers hydrophobic AAs at position +1 and basic AAs at positions -7 to -2. PKCA, like other PKC family members, preferentially phosphorylate peptides with hydrophobic AAs at position +1. It also favors substrates with basic residues at positions -7 to -1, and +2 to +4 [81]. An established consensus substrate motif for PKCA is $(R/K)(R/K)X(S^*/T^*)X(R/K)$ [81].

In contrast to most STKs whose substrate phosphorylation sites are preferentially flanked by basic AA residues, sequences flanking the canonical substrate phosphorylation sites of the protein kinase CK2A1 (catalytic subunit of a constitutively active serine/threonine-protein kinase, Casein Kinase 2 (CK2)) are preferentially occupied by acidic residues [82, 83]. A consensus CK2 substrate motif is $(S^*/T^*)(D/E)X(E/D)$ [83–85]. Acidic residues at positions +1 and +3 are the most important specificity determinants, but additional acidic residues at positions from -2 to +7 and even further contribute to substrate specificity too. Basic and bulky hydrophobic residues at these positions are depleted [82, 83]. Besides local sequence determinants, the overall protein conformation and accessibility of candidate substrate can affect

phosphorylation efficiency of the substrates by CK2 [82]. Additionally, CK2 can use phospho-serine residues as consensus specificity determinants to phosphorylate non-canonical substrates via hierarchical phosphorylation [86].

Using a background set which consisted of randomly sampled peptide sequences of 15 AA residues from the human proteome we reanalyzed the substrate phosphorylation motifs of three human STKs—PKACA, PKCA, and CK2A1—using dagLogo. The substrate sequence patterns identified for PKACA (Fig 3A), PKCA (Fig 3B), and CK2A1 (Fig 3C) by dagLogo are largely consistent with those curated by the PhosphoSitePlus database (https://www.phosphosite.org). Interestingly, we detected significant frequencies of serine residues flanking the phosphorylation sites of CK2A1, where acidic residues are usually preferred. This is consistent with the hierarchical phosphorylation property of CK2 [86]. The substrate motif of PKACA is much more similar to that of PKCA than to that of CK2A1.

Next, we compared the substrate preferences of the three human STKs using all known substrates curated by the PhosphoSitePlus database, with the subsequences of each other's substrates as background sets. The differential phosphorylation site analyses further indicated that the substrate motif of PKACA is much more similar to that of PKCA than to that of CK2A1 (Fig 3). PKCA prefers basic residues at the C-terminal of the phosphorylation sites more than PKACA does, while PKACA prefers basic residues at positions -2 and -3 more than PKAC does, which is evident in Fig 3D. Degenerate logos displaying differential usage of AA residues grouped by charge under physiological conditions further highlight the differential substrate preferences and similarities of the three STKs in terms of charge of residues flanking phosphorylation sites (S2 Fig). The frequencies of Thr and Ser residues at the phosphorylation sites (position 0) of PKCA and PKACA substrates are significantly different (Fig 3D). This example demonstrated that dagLogo enables the discovery of charge preferences around the phosphorylation sites of human STKs with an AA grouping and the differential AA preferences among different kinases.

## Case study 3: dagLogo confirms substrate specificity of the yeast N-terminal acetyltransferase

N-terminal acetylation of proteins by N-terminal acetyltransferases (Nat) is evolutionarily conserved from bacteria to humans [57, 87–89]. More than 80% of human proteins and more than 50% of yeast proteins are N-terminal acetylated [56, 90]. Different N-terminal acetyltransferases (NatA to NatG) have different substrate preferences depending on the N-terminal AA residue context, especially the first and second residues, of mature proteins [57, 88]. About 57% and 84% of N-terminal acetylation of yeast and human proteins are mediated by NatA [56], which preferentially acetylates proteins with Ser-, Ala-, Thr-, Val- or Gly- N-termini [56, 57].

We reanalyzed the yeast NatA substrate specificity using dagLogo with the first 25 residues of yeast NatA substrate as the input set and the first 25 N-terminal residues of yeast proteins excluding the NatA substrates as the background set. Fisher's exact test was performed to determine the significantly differential usage of AAs in the yeast NatA substrate motif. Our analysis demonstrated that the yeast NatA preferentially acetylates proteins with Ser, followed by Ala, as the first residue at the mature N-termini, which is consistent with previous studies [7, 55, 56, 58] (Fig 4A). However, the original publications [56, 57] also reported that proteins with mature N-termini starting with Thr, Val, or Gly residue are preferred by NatA, although no statistical test was performed. In contrast, our analysis does not support such findings (Fig 4A), which is consistent with a reanalysis of the data with a proper background model using iceLogo [7]. Besides the first residues, the residues at positions 2 to 25 also seem to be involved

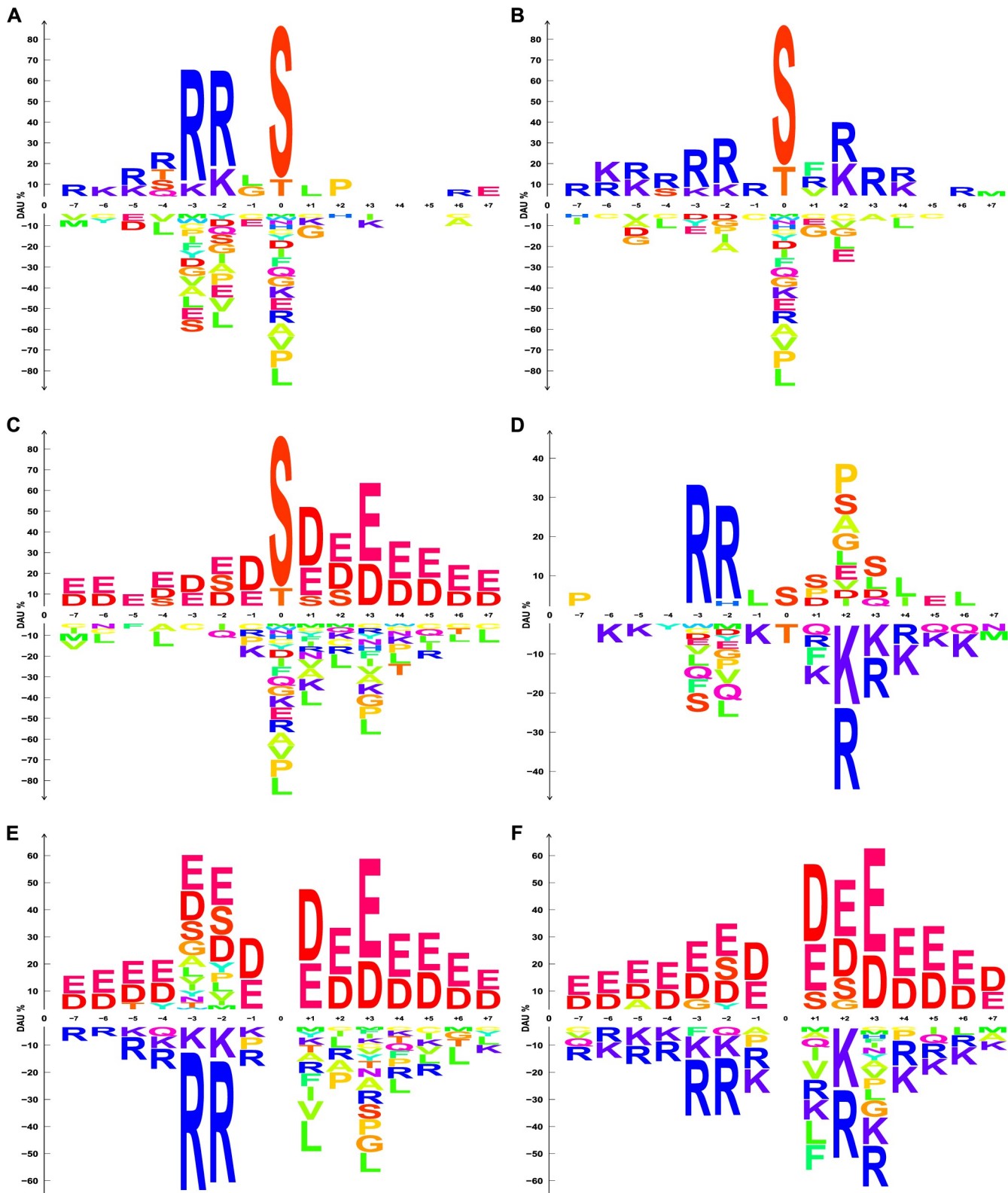

**Fig 3. Sequence logos showing the substrate sequence specificities of human Ser/Thr kinases.** (A-C) Logos showing substrate phosphorylation motifs of human PKACA, PKCA, and CK2A1, respectively. A background model was built from randomly sampled subsequences of 15 AA residues from the UniProt human reference proteome for *Z*-test with a significance level of 0.05. DAU tests were performed to identify AA residue preferences in human substrate

phosphorylation motifs of human PKACA, PKCA, and CK2A1 separately. Over- and under-represented AA residues with significant DAU at each pattern position are shown as sequence logos. Individual AA residues are colored differently. (D-F) Logos showing differential substrate AA residue preferences of PKACA over PKCA, CK2A1 over PKACA, and CK2A1 over PKCA, respectively. Subsequences of 15 AA residues (-7 to +7), centered on the substrate phosphorylation sites of two kinases to compare, were used as the input set and the background set, respectively. Fisher's exact test was performed with a significance level of 0.05.

in determining substrate specificity. Glutamine (Q) residues are preferred at multiple positions of the N-terminal sequences. By grouping AAs based on charge, we also found that yeast NatA strongly prefers uncharged AA residues at position 1 and slightly prefers negatively charged residues at positions 2, 3, 6, 10, 11, 17, 20, and 23, but prefers positively charge residues at position 12 (Fig 4B). Positively charged residues are disliked at positions 1 to 6, and uncharged residues are disfavored at positions 11 and 12 (Fig 4B). When grouping AAs based on size, we observed that yeast NatA strongly prefers tiny residues as the starting AA. In addition, tiny to medium size AAs are favored, and large to gigantic residues are disliked at multiple positions (Fig 4C).

Next, we investigated how different background models affect the results of differential AA usage analysis of the yeast NatA substrate motifs. Background models were built using a background set of the first 25 N-terminal residues of all yeast proteins and a background set of randomly sampled subsequences of 25 AA residues from the yeast reference proteome separately. Fig 4D and 4E show differential AA usage for NatA substrate motifs given the two different background models. It is evident that the identities and percentages of differential usage of some AAs at multiple positions are different when different background models were adopted (Fig 4D and 4E versus Fig 4A). This result underscores the importance of choosing the correct background model based on the study objectives and the advantage of dagLogo.

## Comparison of statistical test methods for differential AA usage analyses

We compared how different statistical test methods, $Z$-test and Fisher's exact test, affect differential AA usage analysis results. For this purpose, we analyzed the human GRB substrate data [53] and the yeast NatA substrate data [56] to represent both ends of the proteome size spectrum with the human proteome (74,823 proteins) as a large and the yeast proteome (6,049 proteins) as a small proteome. Although Fisher's exact test runs slower than z-test for the larger proteome, results based on both methods are largely similar to each other regardless of proteome size (S3 Fig). However, the identities and percentage of differential usage of some AAs are slightly different (S3 Fig).

## Discussion

To facilitate the discovery of conserved patterns from increasingly accumulated proteomic data, we developed the dagLogo package, which have been well downloaded and cited by researchers [4, 80, 91, 92]. Though it does not provide a graphic user interface as other web applications for sequence logo generators (See S1 Table), our package helps to identify and visualize differentially used AA residues or AA residue groups in given protein sequences, improving and extending the sophisticated logo generator iceLogo. First, dagLogo corrects theway of calculating standard deviation of position-specific AA frequencies implemented by iceLogo. In iceLogo, standard deviation of $\hat{p}$ is not correctly computed; it should be $\sqrt{\hat{p}(1-\hat{p})/N}$ instead of $\sqrt{\hat{p}/N}$, where $\hat{p}$ is the sample estimate of the proportion of a given AA residue at a given pattern position, and $N$ is the total number of sequences in a background set [7]. Second, dagLogo implements both $Z$-test and Fisher's exact test, whereas iceLogo only implements $Z$-test. Fisher's exact test computes the exact p-value, while Z-test uses a normal

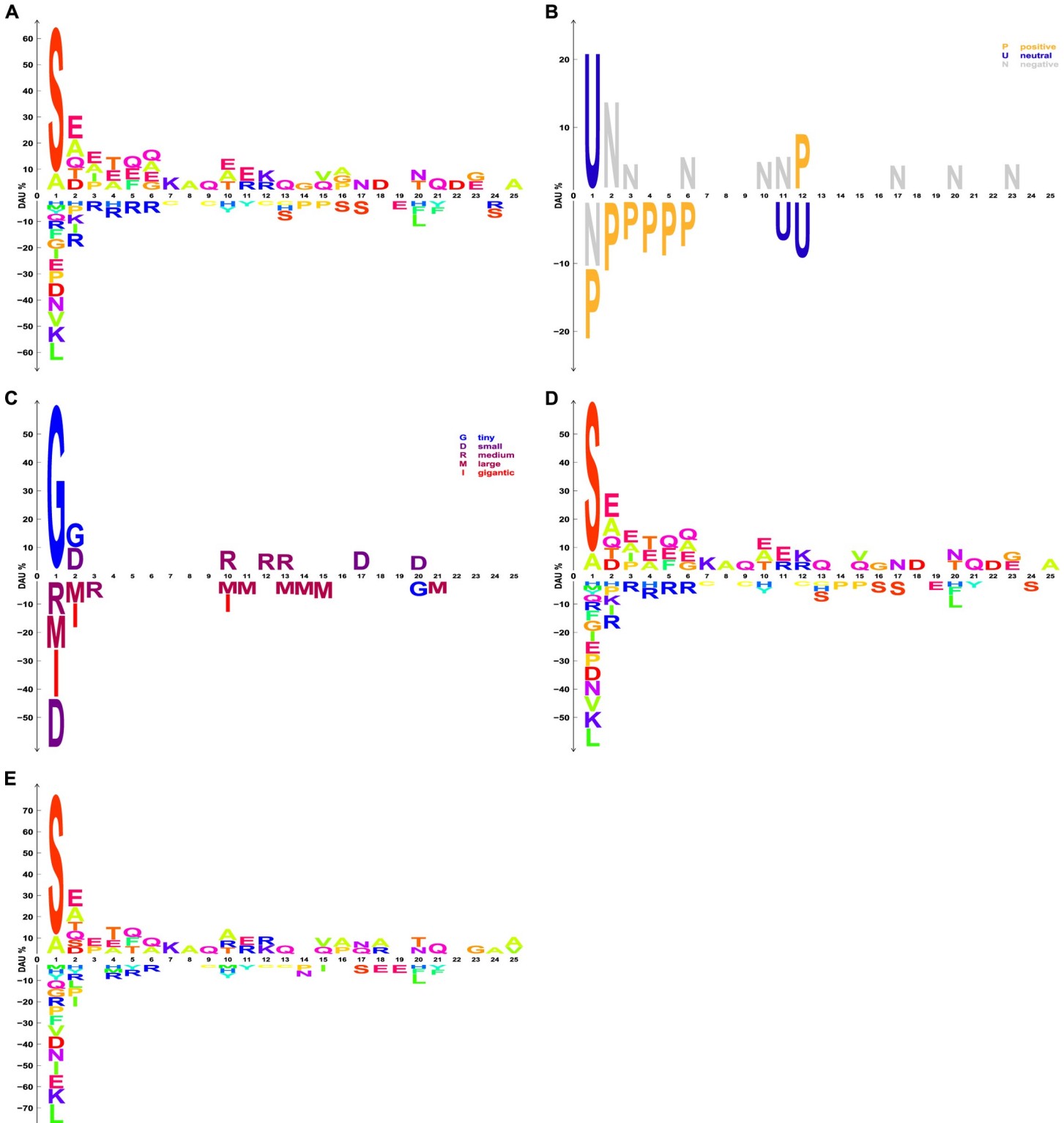

**Fig 4. Sequence logos showing the N-terminal sequence pattern of the yeast NatA substrates.** (A-C) A background model was built from the first 25 N-terminal residues of known yeast, non-NatA substrates for Fisher's exact test with a significance level of 0.05. (A) Over- and under-represented AA residues with significant DAU at each pattern position are shown as sequence logos. Individual AA residues are colored differently. (B and C) AA residues in both the input and background sets are grouped, collapsed, and represented by single letters based on their charge status under physiological conditions and sizes, respectively. Positively charged AAs = {H, K, R}, neutral AAs = {A, C, F, G, I, L, M, N, P, Q, S, T, V, W, Y}, and negatively charged AAs = {D, E}. Tiny = {G, S, A}, small = {D, N, C, E, H}, medium = {R, Q, K, T}, large =

{M, P, Y, F}, and gigantic = {I, L, V, W}. Fisher's exact test was performed with a significance level of 0.05. Over- and under-represented AA residue groups with significant DAU at each pattern position are shown as sequence logos. AA residue groups are colored according to their charge status (B) and sizes (C). (D and E) Background models for Fisher's exact test with a significance level of 0.05 were built from the first 25 N-terminal residues of all yeast proteins and from randomly sampled subsequences of 25 AA residues from the yeast reference proteome, respectively.

distribution as an approximation of a binomial distribution, which becomes problematic when the background sets or input sets are small [93]. When the background sets are large, such as the whole human proteome, both tests produce very similar results, though the Z-test-based method is faster than the Fisher's exact test-based one. However, our analysis also shows that the identities and percentages of differential usage of some AAs can be different (S3 Fig).

Unlike most logo generators, our tool allows multiple different types of inputs. These flexible options can easily eliminate users' efforts to prepare aligned subsequences. Peptide sequences identified by proteomic assays can be directly used as input for pattern discovery. As far as we know, only RNALogo can perform alignment if the input is not pre-aligned [9]. More importantly, twelve background models can be easily generated for Z-test or Fisher's exact test using dagLogo. It is very important to choose an appropriate background model based on the biological questions and experimental protocols. Using the yeast NatA substrate data, we demonstrated the importance of choosing the most appropriate background model (Fig 4A and 4D). Of note, to build the most relevant background model, users may need to focus only on proteins expressed under a given condition, in a specific tissue or cell type, or even in a subcellular compartment.

Additionally, in logos generated by all existing tools, the occurrence of individual residue at a given position is displayed either without considering statistical significance or without allowing degenerate representation based on residue similarity. dagLogo provides more than ten grouping schemes to lump together AA residues of similar physicochemical properties—such as charge, hydrophobicity, and size—into single-letter symbols at given positions. In addition, the function *addScheme* can be used for adding more customized grouping schemes. AA groupings reduce the AA alphabet for testing and visualization. Tests of differential usage of grouped AAs can increase the chances of discovering and visualizing subtle differences that might not be detectable or obvious at the individual AA level (Figs 2, 3 and 4).

## Conclusions

We developed a Bioconductor package, dagLogo, for identifying and visualizing differential AA and AA group usage in proteomics data. Re-analyses of three datasets using dagLogo demonstrated that dagLogo can recapitulate previous findings about the substrate motifs of human granzyme B, three human serine/threonine kinases, and yeast N-terminal acetyltransferase A. Most importantly, analysis with dagLogo at the AA group level revealed important physicochemical property of substrate preferences that would have been hidden with the analysis at the individual AA level alone. The package is well downloaded and already cited in quite a few peer-reviewed journal articles, exemplifying dagLogo as a valuable tool for the analysis of proteomics data. The source code and documentation are freely available at https://bioconductor.org/packages/dagLogo/.

## Supporting information

**S1 Fig. Sequences logos generated by different tools showing sequence determinants of human GRB substrate specificity.** The same set of 416 aligned, equal-length subsequences of 30 residues centered on the cleavage sites was used as the input set. The same set of randomly sampled subsequences of 30 AA residues from the human reference proteome was used as the

background for iceLogo (version 0.2), pLogo (version 1.1.0), and dagLogo (version 1.26.2). A uniform global background of equal probability (0.05) of each of the 20 natural AAs was used for WebLogo (version 2.8.2). (**A**) dagLogo, (**B**) iceLogo, (**C**) pLogo, and (**D**) WebLogo.
(TIF)

**S2 Fig. Degenerate dagLogos showing differential AA group usage in substrate motifs of human STKs.** Subsequences of 15 AA residues (-7 to +7) centered on the substrate phosphorylation sites of two kinases were used as the input set and the background set, respectively. AA residues were grouped by their charge status under physiological conditions, and the differential usage of AA groups was tested by using the function *testDAU* to perform Fisher's exact test with a significance level of 0.05. Positively charged AAs = {H, K, R}, neutral AAs = {A, C, F, G, I, L, M, N, P, Q, S, T, V, W, Y}, and negatively charged AAs = {D, E}. (**A-C**) Logos showing differential substrate AA residue group preferences of PKACA over PKCA, CK2A1 over PKACA, and CK2A1 over PKCA, respectively.
(TIF)

**S3 Fig. dagLogos resulting from different statistical test methods.** Fisher's exact test (**A** and **C**) and *Z*-test (**B** and **D**) were performed to identify substrate sequence preferences of human GRB and yeast NatA, with a significance level of 0.05. For human GRB substrate specificity analyses, the same set of 416, equal-length subsequences of 30 residues centered on the cleavage sites was used as the input set, while background models for Fisher's exact test and *Z*-test were built from all subsequences and from randomly sampled subsequences of 30 AA residues from the UniProt human reference proteome, respectively. For yeast NatA substrate specificity analyses, the same set of subsequences of 25 residues from the N-termini of the 285 NatA substrates was used as the input set, while background models for Fisher's exact test and *Z*-test were built from all subsequences and from randomly sampled subsequences of 25 AA residues from the N-termini of the yeast proteins not including the 285 NatA substrates, respectively. The significance level was set at 0.05 for all tests.
(TIF)

**S1 Table. A comprehensive comparison of existing logo generators.**
(XLSX)

**S2 Table. Description of main functions implemented in the dagLogo package.**
(XLSX)

**S3 Table. Information about substrates of human granzyme B for case study 1.**
(XLSX)

**S4 Table. Information about substrates of human Ser/Thr kinases for case study 2.**
(XLSX)

**S5 Table. Information about substrates of yeast NatA for case study 3.**
(XLSX)

## Acknowledgments

We thank Serena Han at Harvard College for editorial assistance.

## Author Contributions

**Conceptualization:** Lihua Julie Zhu.

**Data curation:** Haibo Liu, Usha Acharya.

**Formal analysis:** Haibo Liu, Lihua Julie Zhu.

**Investigation:** Jianhong Ou, Haibo Liu, Lihua Julie Zhu.

**Methodology:** Jianhong Ou, Haibo Liu, Alexey Stukalov, Lihua Julie Zhu.

**Project administration:** Lihua Julie Zhu.

**Resources:** Michael R. Green, Lihua Julie Zhu.

**Software:** Jianhong Ou, Haibo Liu, Alexey Stukalov.

**Supervision:** Lihua Julie Zhu.

**Validation:** Jianhong Ou, Haibo Liu, Niraj K. Nirala, Usha Acharya, Michael R. Green.

**Visualization:** Jianhong Ou, Haibo Liu.

**Writing – original draft:** Haibo Liu, Lihua Julie Zhu.

**Writing – review & editing:** Jianhong Ou, Haibo Liu, Niraj K. Nirala, Alexey Stukalov, Usha Acharya, Michael R. Green, Lihua Julie Zhu.

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
