## [Decision Letter · Decision Letter 0]

19 Oct 2020

PONE-D-20-29957

dagLogo: an R/Bioconductor package for identifying and visualizing differential amino acid group usage in proteomics data

PLOS ONE

Dear Dr.Zhu,

Thank you for submitting your manuscript to PLOS ONE. After careful consideration, we feel that it has merit but does not fully meet PLOS ONE’s publication criteria as it currently stands. Therefore, we invite you to submit a revised version of the manuscript that addresses the points raised during the review process.

This version of the manuscript is lacking some explanations regarding the sustainability of the project as well as its comparability with other similar tools.

Please submit your revised manuscript by November 30,2020 If you will need more time than this to complete your revisions, please reply to this message or contact the journal office at plosone@plos.org. Please include the following items when submitting your revised manuscript:

We look forward to receiving your revised manuscript.

Kind regards,

Frederique Lisacek

Academic Editor

PLOS ONE

Journal Requirements:

Reviewers' comments:

Reviewer's Responses to Questions

**Comments to the Author**

1. Is the manuscript technically sound, and do the data support the conclusions?

Reviewer #1: Yes

Reviewer #2: Yes

2. Has the statistical analysis been performed appropriately and rigorously? 

Reviewer #1: N/A

Reviewer #2: Yes

3. Have the authors made all data underlying the findings in their manuscript fully available?

Reviewer #1: Yes

Reviewer #2: Yes

4. Is the manuscript presented in an intelligible fashion and written in standard English?

Reviewer #1: Yes

Reviewer #2: Yes

5. Review Comments to the Author

Reviewer #1: In this manuscript the authors present the dagLogo R package, a software for identifying and visualizing differential amino acids in protein sequences. Citation of previous related work was properly made; the algorithm and R implementation are well described, and utilization of the tool was properly evaluated with publicly available data. The package could be installed and tested.

The following suggestions are made to improve the manuscript.

1. In the introduction the several advantages of dagLogo over other tools are described. While an R package certainly provides more flexibility, it would be fair to mention that other tools provide graphical user interfaces which are easier to use.

2. The structure or order of sections should be revised. For example, Materials and Methods should be moved up and Discussion should be the last one.

3. The resolution of the images is not good enough to see details.

4. Line 99, “AAs has been”  “AAs have been”.

5. Line 159, definition of “DAU”. This was defined later with the current structure, it could be addressed if the sections are re-organized, as mentioned previously.

6. Line 173, “Van Damme et al. thinks that”. Use a more formal term, e.g. “Van Damme et al. propose that”.

Reviewer #2: Ou et al. present the dagLogo R package to generate sequence logos from protein and peptide input. Some of the major features of dagLogo are its support of custom background models for differential amino acid analysis and the ability to group amino acids based on physicochemical properties.

The article is well written and contains relevant examples.

It is mentioned that a disadvantage of existing tools is that they do not provide alignment functionality (page 3 line 72). However, I could not find a description of alignment functionality built into dagLogo either; instead the examples always use aligned sequences of a fixed length (optionally padded). Can it be made clearer whether alignment is available within dagLogo, and if this is not the case, can the authors discuss the lack of this functionality?

Additionally I am wondering how well dagLogo is supported. Although it is a project with proven longevity (available since 2013), there is an open issue on GitHub from 2018 that has not yet been addressed. Furthermore, the latest available version on Bioconductor (1.26.2) seems to lag behind the last GitHub version (1.27.9).

Minor comments:

- Page 11 line 233: Highlight the modified S/T residues similarly as on lines 227 and 239.

- Page 21 line 444: The acronym DAU is only defined here, whereas it is repeatedly used in the preceding text. Please define this acronym upon first use.

- Page 22 line 475: "Uniport" -> UniProt

- Page 22 line 476: How does dagLogo determine which characteristics are relevant to fetch similar sequences?

- Information on data and software versions is largely missing. For example, what is the version of the UniProt and SwissProt proteomes that were used, or the versions of the logo generators against which was compared in figure S1, etc.?

- The figures are of low quality: they are pixelated and some of the text is hard to read. Potentially this is an artefact of the upload procedure, but if not the figures should be exported in a higher quality.

6. PLOS authors have the option to publish the peer review history of their article (what does this mean?). If published, this will include your full peer review and any attached files.

Reviewer #1: **Yes: **Aivett Bilbao

Reviewer #2: No

---

## [Author Response · Author response to Decision Letter 0]

22 Oct 2020

We greatly appreciate the time and effort given by the reviewers to thoughtfully read our manuscript and test our software. We thank the reviewers for the positive feedback and great suggestions! 

We have addressed all of the comments below each one.

Reviewers' comments:

Reviewer's Responses to Questions

Comments to the Author

1. Is the manuscript technically sound, and do the data support the conclusions?

Reviewer #1: Yes

Reviewer #2: Yes

2. Has the statistical analysis been performed appropriately and rigorously?

Reviewer #1: N/A

Reviewer #2: Yes

3. Have the authors made all data underlying the findings in their manuscript fully available?

Reviewer #1: Yes

Reviewer #2: Yes

4. Is the manuscript presented in an intelligible fashion and written in standard English?

Reviewer #1: Yes

Reviewer #2: Yes

5. Review Comments to the Author

Reviewer #1: In this manuscript the authors present the dagLogo R package, a software for identifying and visualizing differential amino acids in protein sequences. Citation of previous related work was properly made; the algorithm and R implementation are well described, and utilization of the tool was properly evaluated with publicly available data. The package could be installed and tested.

The following suggestions are made to improve the manuscript.

1. In the introduction the several advantages of dagLogo over other tools are described. While an R package certainly provides more flexibility, it would be fair to mention that other tools provide graphical user interfaces which are easier to use.

Response: We thank the reviewer for the great suggestion! We added this in the Discussion section of the revised manuscript as tracked.

2. The structure or order of sections should be revised. For example, Materials and Methods should be moved up and Discussion should be the last one.

Response: We reordered the sections per the reviewer’s request.

3. The resolution of the images is not good enough to see details.

Response: We have generated high-quality figures as suggested. We notice that the figure quality is low in the auto-generated pdf file with all files together. We suggest the reviewer download individual TIFF files to obtain high resolution figures.

4. Line 99, “AAs has been”  “AAs have been”.

Response: We thank the reviewer for spotting this error which we have fixed.

5. Line 159, definition of “DAU”. This was defined later with the current structure, it could be addressed if the sections are re-organized, as mentioned previously.

Response: As the reviewer suspected, this has been addressed after we reorganized the sections per the reviewer’s suggestion. 

6. Line 173, “Van Damme et al. thinks that”. Use a more formal term, e.g. “Van Damme et al. propose that”.

Response: We thank the reviewer for a better word choice. We changed “thinks” to “proposed” in the revised manuscript.

Reviewer #2: Ou et al. present the dagLogo R package to generate sequence logos from protein and peptide input. Some of the major features of dagLogo are its support of custom background models for differential amino acid analysis and the ability to group amino acids based on physicochemical properties.

The article is well written and contains relevant examples.

It is mentioned that a disadvantage of existing tools is that they do not provide alignment functionality (page 3 line 72). However, I could not find a description of alignment functionality built into dagLogo either; instead the examples always use aligned sequences of a fixed length (optionally padded). Can it be made clearer whether alignment is available within dagLogo, and if this is not the case, can the authors discuss the lack of this functionality?

Response: dagLogo accepts aligned subsequences as well as unaligned subsequences of different lengths as input. With unaligned input sequences, dagLogo align them using the input positions or identities of anchoring amino acid (AA), and the upstream and downstream offsets relative to the anchoring AA. For examples with unaligned input sequences, please type help(fetchSequence) or refer to https://rdrr.io/bioc/dagLogo/man/fetchSequence.html. 

To make it clearer, we have added more descriptions in line 165 – 166.

Additionally, I am wondering how well dagLogo is supported. Although it is a project with proven longevity (available since 2013), there is an open issue on GitHub from 2018 that has not yet been addressed. Furthermore, the latest available version on Bioconductor (1.26.2) seems to lag behind the last GitHub version (1.27.9).

Response: We thank the reviewer for bringing our attention to the posted issue. We have posted our response to the issue. There is a central support site https://support.bioconductor.org for all Bioconductor packages. We have been closely monitoring and actively supporting a dozen of our Bioconductor packages through this site. In fact, we have posted more than 5400 messages at https://support.bioconductor.org/user/list/?sort=reputation&limit=all%20time&q=. In the future, we will periodically check for issues in the GitHub as well.

Regarding the fact that the latest available version on Bioconductor (1.26.2) seems to lag behind the last GitHub version (1.27.9), this is usually the case because the GitHub version is the development version while the Bioconductor version 1.26.2 is the released version. In fact, Bioconductor hosts development version as well at http://bioconductor.org/packages/devel/bioc/html/dagLogo.html, which should be consistent with the GitHub version. Bioconductor has two releases each year, one in April and the other in October. Once a version has been released, only bug fixes and documentation improvements can be made. Any other types of improvements can only be made to the development branch and be included in the subsequent releases.

Minor comments:

- Page 11 line 233: Highlight the modified S/T residues similarly as on lines 227 and 239.

Response: We thank the reviewer for pointing this out. We highlighted all the S/T residues to make the style consistent between lines.

- Page 21 line 444: The acronym DAU is only defined here, whereas it is repeatedly used in the preceding text. Please define this acronym upon first use.

Response: We reorganized the sections per the other reviewer’s recommendation. DAU is now defined in the Materials and Methods section at its first appearance.

- Page 22 line 475: "Uniport" -> UniProt

Response: We thank the reviewer for spotting the error which we have corrected.

- Page 22 line 476: How does dagLogo determine which characteristics are relevant to fetch similar sequences?

Response: The fetchSequence() function in the dagLogo package provide parameters for user to specify the identity of anchoring amino acid, upstream and downstream offsets relative to the anchoring amino acids. By setting these parameters, dagLogo can fetch subsequences containing anchoring amino acids at the specified positions.

- Information on data and software versions is largely missing. For example, what is the version of the UniProt and SwissProt proteomes that were used, or the versions of the logo generators against which was compared in figure S1, etc.?

Response: We thank the reviewer for the advice! We have added the version of the database (Materials and Methods) and software (Fig S1. Legend) in the revised manuscript.

- The figures are of low quality: they are pixelated and some of the text is hard to read. Potentially this is an artefact of the upload procedure, but if not the figures should be exported in a higher quality.

Response: We regenerated all figures in high-quality TIFF format. We notice that the figure quality is low in the auto-generated pdf file with all files together. We suggest the reviewer download individual TIFF files to obtain high resolution figures.

---

## [Editor Report · Decision Letter 1]

26 Oct 2020

dagLogo: an R/Bioconductor package for identifying and visualizing differential amino acid group usage in proteomics data

PONE-D-20-29957R1

Dear Dr. Zhu,

We’re pleased to inform you that your manuscript has been judged scientifically suitable for publication and will be formally accepted for publication once it meets all outstanding technical requirements.

Kind regards,

Frederique Lisacek

Academic Editor

PLOS ONE
---

## [Editor Report · Acceptance letter]

28 Oct 2020

PONE-D-20-29957R1 

dagLogo: an R/Bioconductor package for identifying and visualizing differential amino acid group usage in proteomics data 

Dear Dr. Zhu:

I'm pleased to inform you that your manuscript has been deemed suitable for publication in PLOS ONE. Congratulations! Your manuscript is now with our production department. 

Kind regards, 

on behalf of

Dr. Frederique Lisacek 

Academic Editor

PLOS ONE